# A Statistical Recurrent Model on the Manifold of Symmetric Positive Definite Matrices[*]

**Rudrasis Chakraborty**[†]    **Chun-Hao Yang**[†♯]    **Xingjian Zhen**[‡♯]    **Monami Banerjee**[†]

**Derek Archer**[†]    **David Vaillancourt**[†]    **Vikas Singh**[‡]    **Baba C. Vemuri**[†]

[†]University of Florida, Gainesville, USA
[‡]University of Wisconsin Madison, USA

[♯] Equal contribution

## Abstract

In a number of disciplines, the data (e.g., graphs, manifolds) to be analyzed are non-Euclidean in nature. Geometric deep learning corresponds to techniques that generalize deep neural network models to such non-Euclidean spaces. Several recent papers have shown how convolutional neural networks (CNNs) can be extended to learn with graph-based data. In this work, we study the setting where the data (or measurements) are ordered, longitudinal or temporal in nature and live on a Riemannian manifold – this setting is common in a variety of problems in statistical machine learning, vision and medical imaging. We show how recurrent statistical network models can be defined in such spaces. Then, we present an efficient algorithm and conduct a rigorous analysis of its statistical properties. We perform numerical experiments demonstrating competitive performance with state of the art methods but with significantly fewer parameters. We also show applications to a statistical analysis task in brain imaging, a regime where deep neural network models have only been utilized in limited ways.

## 1   Introduction

In the last decade or so, deep neural network models have been very successful in learning complicated patterns from data such as images, videos and speech [41, 39] – this has led to a number of breakthroughs as well as deployments in turnkey applications. A popular neural network architecture that has contributed to these advancements is convolutional neural networks (CNNs). In the classical definition of convolution, one often assumes that the data correspond to discrete measurements, acquired at equally spaced intervals (i.e., Euclidean space), of a scalar (or vector) valued function. Clearly, for images, the Euclidean lattice grid assumption makes sense and the use of convolutional architectures is appropriate – as described in [11], a number of properties such as stationarity, locality and compositionality follow. While the assumption that the underlying data satisfies the Euclidean structure is explicit or implicit in an overwhelming majority of models, recently there has been a growing interest in applying or extending deep learning models for non-Euclidean data. This line of work is called *Geometric deep learning* and typically deals with data such as manifolds and graphs [11]. Existing results describe strategies for leveraging the mathematical properties of such geometric or structured data, specifically, lack of  **(a)** global linear structure, **(b)** global coordinate system, **(c)** shift invariance/equivariance, by incorporating these ideas explicitly into deep networks used to model them [13, 37, 18, 31, 30, 19].

Separate from the evolving body of work at the interface of convolutional neural networks and structured data, there is a mature literature in statistical machine learning [40] and computer vision

---

[*]This research was funded in part by the NSF grant IIS-1525431 and IIS-1724174 to BCV, R01 NS052318 to DV and NSF CAREER award 1252725 and R01 EB022883 to VS. XZ and VS were also supported by UW CPCP (U54 AI117924).

demonstrating how exploiting the *structure* (or geometry) of the data can yield advantages. Structured data abound in various data analysis tasks: directional data in measurements from antennas [44], time series data (curves) in finance [60] and health sciences [20], surface normal vectors on the unit sphere (in vision or graphics) [58], probability density functions (in functional data analysis) [56], covariance matrices (for use in conditional independences, image textures) [62], rigid motions (registration) [48], shape representations (shape space analysis) [34], tree-based data (parse trees in natural language processing) [51], subspaces (videos, segmentation) [65, 23], low-rank matrices [12, 63], and kernel matrices [53] are common examples. In neuroimaging, an image may have a structured measurement at each voxel to describe water diffusion [7, 64, 42, 32, 4, 15, 35] or local structural change [29, 68, 36]. And the study of the interface between geometry/structure and analysis methods has given effective practical tools because in order to define loss functions that make sense for the data at hand, one needs to first define a metric which is intrinsic to the structure of the data.

The foregoing discussion, for the most part, covers differential geometry inspired algorithms for *non-sequential* (or non-temporal) data. The study of analogous schemes for temporal or longitudinal data is less well-developed. But analysis of dynamical scenes and stochastic processes is an important area of machine learning and vision, and it is here that some results have shown the benefits of explicitly using geometric ideas. Some of the examples include the modeling of temporal evolution of features in dynamic scenes in action recognition [2, 9, 61], tractography [14, 50] and so on. There are also proposals describing modeling stochastic linear dynamical system (LDS) [22, 2, 9, 61]. In [2, 3], authors studied the Riemannian geometry of LDS to define distances and first order statistics. Given that the marriage between deep learning and learning on non-Euclidean domains is a fairly recent, the existing body of work primarily deals with attempts to generalize the popular CNN architectures. Few results exist that study *recurrent* models for non-Euclidean structured domains.

The broad success of Recurrent Neural Network (RNN) architectures including Long short term memory (LSTM) [28] and Gated recurrent unit (GRU) [17] in sequential modeling like Natural Language Processing (NLP) has motivated a number of attempts to apply such ideas to model stochastic processes or to characterize dynamical scenes which can be viewed as a sequence of images. Several works have proposed variants of RNN to model dynamical scenes including [57, 21, 46, 54, 66]. In the recent past, developments have been made to reduce the number of parameters in RNN and making RNN faster [38, 66]. In [6, 27], authors proposed an efficient way to handle vanishing and exploding gradient problem of RNN using unitary weight matrices. In [33], authors proposed a RNN model which combines the remembering ability of unitary RNNs with the ability of gated RNNs to effectively forget redundant/ irrelevant information. Despite these results, we find that no existing model describes a recurrent model for structured (specifically, manifold-valued) data. The **main contribution** of this paper is to describe a recurrent model (and accompanying theoretical analysis) that will fall under the umbrella of "geometric deep learning" — it exploits the geometry of non-Euclidean data but is specifically designed for temporal or ordered measurements.

## 2 Preliminaries: Key Ingredients from Riemannian geometry

In this section, we will first give a brief overview of the Riemannian geometry of $n \times n$ symmetric positive definite matrices (henceforth will be denoted by $\mathsf{SPD}(n)$). Note that our development is not limited to $\mathsf{SPD}(n)$, but choosing a specific manifold simplifies the presentation and the notations significantly. Then, we will present key ingredients needed for our proposed recurrent model.

**Differential Geometry of $\mathsf{SPD}(n)$:** Let $\mathsf{SPD}(n)$ be the set of $n \times n$ symmetric positive definite matrices. The group of $n \times n$ full rank matrices, denoted by $\mathsf{GL}(n)$ and called the general linear group, acts on $\mathsf{SPD}(n)$ via the group action, $g.A := gAg^T$, where $g \in \mathsf{GL}(n)$ and $A \in \mathsf{SPD}(n)$. One can define a $\mathsf{GL}(n)$ invariant intrinsic metric, $d_{\mathsf{GL}}$ on $\mathsf{SPD}(n)$ as $d_{\mathsf{GL}}(A, B) = \sqrt{\mathsf{trace}(\mathsf{Log}(A^{-1}B)^2)}$, see [26]. Here, $\mathsf{Log}$ is the matrix logarithm. This metric is intrinsic but requires a spectral decomposition for calculations, a computationally intensive task for large matrices. In [16], the Jensen-Bregman LogDet (JBLD) divergence was introduced on $\mathsf{SPD}(n)$. As the name suggests, this is not a metric but as proved in [55], the square root of JBLD turns out to be a metric (called the Stein metric), which is defined as $d(A, B) = \sqrt{\log \det(\frac{A+B}{2}) - \frac{1}{2} \log \det(AB)}$.

Here, we used the notation $d$ without any subscript to denote the Stein metric. It is easy to see that the Stein metric is computationally much more efficient than the $\mathsf{GL}(n)$-invariant natural metric on $\mathsf{SPD}(n)$ as no eigen decomposition is required. This will be useful for training our recurrent model. In the remainder of the paper, we will assume the metric on $\mathsf{SPD}(n)$ to be the Stein metric. Now, we describe a few operations on $\mathsf{SPD}(n)$ which are needed to define the recurrent model.

**"Translation" operation on SPD$(n)$:** Let $I$ be the set of all isometries on SPD$(n)$, i.e., given $g \in I$, $d(g.A, g.B) = d(A, B)$, for all $A, B \in$ SPD$(n)$, where . is the group action as defined earlier. It is clear that $I$ forms a group (henceforth, will be denoted by $G$) and for a given $g \in G$ and $A \in$ SPD$(n)$, $g.A \mapsto B$, for some $B \in$ SPD$(n)$ is a group action. One can easily see that, endowed with the Stein metric, $G =$ GL$(n)$. In this work, we will choose a subgroup of GL$(n)$, i.e., O$(n)$ as our choice of $G$, where, O$(n)$ is the set of $n \times n$ orthogonal matrices and $g.A := gAg^T$. Since the O$(n)$ group operation preserves the distance, we call this group operation "translation", analogous to the case of Euclidean space and is denoted by $\mathsf{T}_A(g) := gAg^T$.

**Parametrization of SPD$(n)$:** Let $A \in$ SPD$(n)$. We will obtain the Cholesky factorization of $A = LL^T$, where $L$ is an invertible lower traingular matrix. This gives a unique parametrization of SPD$(n)$. Let the parametrization be $A = \mathsf{Chol}((l_1, l_2, \cdots l_n, \cdots, l_{n(n+1)/2})^t)$. With a slight abuse of notation, we will use Chol to denote both decomposition and construction based on the type of the domain of the function, i.e., $\mathsf{Chol}(A) := L$ and $\mathsf{Chol}(L) := LL^T = A$. Note that here $l_1, l_2, \cdots, l_n$ are diagonal entries of $L$ and are positive and $l_{n+1}, \cdots, l_{n(n+1)/2}$ can be any real numbers.

**Parametrization of O$(n)$:** O$(n)$ is a Lie group [25] of $n \times n$ orthogonal matrices (of dimension $n(n-1)/2$) with the corresponding Lie algebra, $\mathfrak{O}(n)$, and consists of the set of $n \times n$ skew-symmetric matrices. The Lie algebra is a vector space, so we will use the corresponding element from the Lie algebra to parametrize a point on O$(n)$. Let $g \in$ O$(n)$, we will use the matrix logarithm of $\mathfrak{g} = \log(g)$ to get the parametrization as the skew-symmetric matrix. So, $g = \exp((\mathfrak{g}_1, \mathfrak{g}_2, \cdots, \mathfrak{g}_{n(n-1)/2})^t)$. exp is the matrix exponential operator.

**Weighted Fréchet mean (wFM) of matrices on SPD$(n)$:** Given $\{X_i\}_{i=1}^N \subset$ SPD$(n)$, and $\{w_i\}_{i=1}^N$ with $w_i \geq 0$, for all $i$ and $\sum_i w_i = 1$, the weighted Fréchet mean (wFM) [24] is:

$$M^* = \operatorname*{argmin}_M \sum_{i=1}^N w_i d^2(X_i, M) \tag{1}$$

The existence and uniqueness of the Fréchet mean (FM) is discussed in detail in [1]. In this paper, we will assume that the samples lie within a geodesic ball of an appropriate radius so that FM exists and is unique. We will use $\mathsf{FM}(\{X_i\}, \{w_i\})$ to denote the wFM of $\{X_i\}$ with weights $\{w_i\}$. With the above tools in hand, now we are ready to formulate the Statistical Recurrent Neural Network on SPD$(n)$, dubbed as SPD-SRU.

## 3   A Statistical Recurrent Network Model in the space of SPD$(n)$ matrices

The main motivation for our work comes from the statistical recurrent unit (SRU) model on Euclidean spaces in [47]. To setup our formulation, we will briefly review the SRU formulation followed by details of our recurrent model for manifold valued measurements.

**What is the Statistical Recurrent Unit (SRU)?**  The authors in [47] propose an interesting model for sequential (or temporal) data based on an un-gated recurrent unit (called Statistical Recurrent Unit (SRU)). The model maintains the sequential dependency in the input samples through a simple summary statistic — the so-called *exponential moving average*. Even though the proposal is based on an un-gated architecture, the development and experiments show that the results from SRU are competitive with more complex alternatives like LSTM and GRU. One reason put forth in that work is that using appropriately designed summary statistics, one can essentially emulate complicated gated units and still capture long terms relations (or memory) in sequences. This property is particularly attractive when we study recurrent models for more complicated measurements such as manifolds. Recall that the key challenge in extending statistical machine learning models to manifolds involves re-deriving many of the classical (Euclidean) arithmetic and geometric operations while respecting the geometry of the manifold of interest. The simplicity of un-gated units provides an excellent starting point. Below, we describe the key update equations that define the SRU.

Let $\mathbf{x}_1, \mathbf{x}_2, \cdots \mathbf{x}_T$ be an input sequence on $\mathbf{R}^n$, presented to the model. As in most recurrent models, the training process in SRU proceeds by updating the weights of the model. Let the weight matrix be denoted by $W$ (the node is indexed by the superscript). The update rules for SRU are as follows:

$$\mathbf{r}_t = \mathrm{ReLU}\left(W^{(r)}\boldsymbol{\mu}_{t-1} + b^{(r)}\right) \tag{2}$$

$$\forall \alpha \in J\,,\quad \boldsymbol{\mu}_t^{(\alpha)} = \alpha\boldsymbol{\mu}_{t-1}^{(\alpha)} + (1-\alpha)\boldsymbol{\varphi}_t \tag{4}$$

$$\boldsymbol{\varphi}_t = \mathrm{ReLU}\left(W^{(\phi)}\mathbf{r}_t + W^{(x)}\mathbf{x}_t + b^{(\phi)}\right) \tag{3}$$

$$\mathbf{o}_t = \mathrm{ReLU}\left(W^{(o)}\boldsymbol{\mu}_t + b^{(o)}\right) \tag{5}$$

where $J$ is the set of different scales. The SRU formulation is analogous to mean map embedding (MME) but applied to non i.i.d. samples. Since the average of a set of i.i.d. samples will essentially

marginalize over time, simple averaging will lose the temporal/sequential information. On the other hand, the SRU computes a moving average over time which captures the average of the data seen so far, i.e., when computing $\mu$ from $\varphi$ (as shown in Fig. 1). This is very similar to taking the average of stochastic processes and looking at the "average process". Further, by looking at averages over different scales, essentially, we can uncover statistics computed over different time scales. This is because $\mu$ is not only a function of $\phi$ but also a function of $\{\mathbf{x}_i\}_{i=1}^{t-1}$ via $\mathbf{r}_t$. This dependence on the past "tokens" in the sequence is shown in Fig. 1 by a "dashed" line. With this description, we can easily list the key operational components in the update rules in (2)-(5) and then evaluate if such components can be generalized to serve as the building blocks of our proposed model.

**Which low-level operations are needed?** We can verify that the key ingredients to define the model in SRU are **(i)** weighted sum; **(ii)** addition of bias; **(iii)** moving average and **(iv)** non-linearity. In principle, if we can generalize each of these operations to the $\mathsf{SPD}(n)$ manifold, it will provide us the basic components to define the model. Observe that items (i) and (iii) are essentially a **weighted sum** if we impose a convexity constraint on the weights. Then, the weighted sum for the Euclidean setting can be generalized using wFM as defined in Section 2 (denoted by $\mathsf{FM}$).

If we can do so, it will also provide a way to compute moving averages on $\mathsf{SPD}(n)$. Now, the second operation we can identify above is the **translation** on Euclidean spaces. This can be achieved by the "translation" operation on $\mathsf{SPD}(n)$ as defined in Section 2 (denoted by $\mathsf{T}$). Finally, in order to generalize ReLU on $\mathsf{SPD}(n)$, we will use the standard ReLU on the parameter space (this will be the local *chart* of $\mathsf{SPD}(n)$) and then map it back on to the manifold. This means that we have generalized each of the key components. With this in hand, we are ready to present our proposed recurrent model on $\mathsf{SPD}(n)$. We first formally describe our

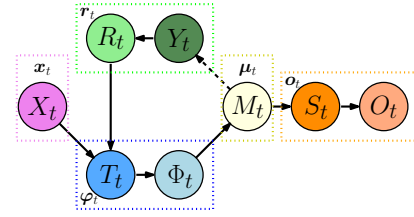

Figure 1: Sketch of an SPD-SRU and SRU layer (dashed line represnets dependence on the previous time point).

SPD-SRU layer and then contrast with the SRU layer, to help see the main differences.

**Basic components of the SPD-SRU model.** Let, $X_1, X_2, \cdots X_T$ be an input temporal or ordered sequence of points on $\mathsf{SPD}(n)$. The update rules for a layer of SPD-SRU is as follows:

$$Y_t = \mathsf{FM}\left(\left\{M_{t-1}^{(\alpha)}\right\}, \left\{w^{(y,\alpha)}\right\}\right), \quad R_t = \mathsf{T}\left(Y_t, g^{(r)}\right) \tag{6}$$

$$T_t = \mathsf{FM}\left(\{R_t, X_t\}, w^{(t)}\right), \quad \Phi_t = \mathsf{T}\left(T_t, g^{(p)}\right) \tag{7}$$

$$\forall \alpha \in J, \quad M_t^{(\alpha)} = \mathsf{FM}\left(\left\{M_{t-1}^{(\alpha)}, \Phi_t\right\}, \alpha\right) \tag{8}$$

$$S_t = \mathsf{FM}\left(\left\{M_t^{(\alpha)}\right\}, \left\{w^{(s,\alpha)}\right\}\right), \quad O_t = \mathsf{Chol}\left(\mathsf{ReLU}\left(\mathsf{Chol}\left(\mathsf{T}\left(S_t, g^{(y)}\right)\right)\right)\right) \tag{9}$$

where, $t \in \{1, \cdots, T\}$ and $M_0^{(\alpha)}$ is initialized to be a diagonal $n \times n$ matrix with small positive values. Similar to before, the set $J$ consists of positive real numbers from the unit interval. Now, computing the FM at the different elements of $J$ will give a wFM at different "scales", exactly as desired. Analogous to the SRU, here $M_t^{(\alpha)}$s are computed by averaging $\Phi_t$ at different scales as shown in Fig. 1. This model leverages the context based on previous data by asking the moving averages, $M_t^{(\alpha)}$ to depend on past data, $\{X_i\}_{i=1}^{t-1}$ through $R_t$ (as shown in Fig. 1).

**Comparison between the SPD-SRU and the SRU layer:** In the SPD-SRU unit above, each update identity is a generalization of an update equation of SRU. In (6), we compute the weighted combination of the previous FMs (computed using different "scales") with a "translation", i.e., the input is $\{M_{t-1}^{(\alpha)}\}$ and the output is $R_t$. This update equation is analogous to the weighted combination of the past means with bias as given in (2)) where the input is $\{\mu_{t-1}^{(\alpha)}\}$ and the output is $\mathbf{r}_t$. This update rule calculates a weighted combination of the past information. In (7), we compute a weighted combination of the previous information, $R_t$ and the current point or token, $X_t$ with a "translation". The input of this equation is $R_t$ and $X_t$ and the output is $\Phi_t$. This is analogous to (3), where the input is $\mathbf{r}_t$ and $\mathbf{x}_t$ and the output is $\varphi_t$. This update rule combines old and new information. Now, we will update the new information based on the combined information at the current time step, i.e., $\Phi_t$. This is accomplished in (8). Here, we are computing an FM (average) at different "scales". Computing averages at different "scales" essentially allows including information from previous data points which have been seen at various time scales. This step is a generalization of (4). In this step, the input is $\left\{M_t^{(\alpha)}\right\}$ and $\Phi_t$ (with $\{\mu_{t-1}^{(\alpha)}\}$ and $\varphi_t$ respectively) and the output is $\{M_t^{(\alpha)}\}$ (with $\{\mu_t^{(\alpha)}\}$).

This step is the combined information gathered at the current time step. Finally, in (9), we used a weighted combination of the current FMs (averages) and outputs $O_t$. This is the last update rule in SRU, i.e., (5). Observe that we did *not* use the ReLU operation in each update rule of SPD-SRU, in contrast to SRU. This is because, these update rules are highly nonlinear unlike in the SRU, hence, a ReLU unit at the final output of the layer is sufficient. Also, notice that $O_t \in \mathsf{SPD}(n)$, hence, we can cascade multiple SPD-SRU layers, in other words in the next layer, the input sequence will be $O_1, O_2 \cdots O_T$. The update equations track the "averages" (FM) at varying scales. This is the reason we can call our framework statistical recurrent network. We will shortly see that our framework can utilize parameters more efficiently and requires very few parameters because of the ability to use the covariance structure.

**Important properties of SPD-SRU model:** The "translation" operator $\mathsf{T}$ is analogous to "adding" a bias term in a standard neural network. One reason we call it "translation" is because the action of $\mathsf{O}(n)$, preserves the metric. Notice that although in this description, we track the FMs at different scales, one may easily use other statistics, e.g., Fréchet median and mode, etc. The key bottleneck is to efficiently compute the moving statistic (whatever it may be), which will be discussed shortly. Note that the SPD-SRU formulation can be generalized to other manifolds. In fact, it can be easily generalized to Riemannian homogeneous spaces [26] because of two reasons **(a)** closed form expressions for Riemannian exponential and inverse exponential maps exist and **(b)** a group $G$ acts transitively on these spaces, hence we can generalize the definition of "translation". Other manifolds are also possible but the technical details will be different. Now, we will comment on learning the parameters of our proposed model.

**Learning the parameters:** Notice that using the parametrization of $\mathsf{O}(n)$, we will learn the "bias" term on the parametric space, which is a vector space. The weights in the wFM must satisfy the non-negativity constraint. In order to ensure that this property is satisfied, we will learn the square root of the weights which is unconstrained, i.e., the entire real line. We will impose the affine constraint explicitly by normalizing the weights. Hence, all the trainable parameters lie in the Euclidean space and the optimization of these parameters is unconstrained, hence standard techniques are sufficient.

**Remarks.** It is interesting to observe that the update equations in (6)-(9) involve group operations and wFM computation. But as evident from the (1), the wFM computation requires numerical optimization, which is computationally *not* efficient. This is a bottleneck. For example, for our proposed model, on a batch size of 20 with $15 \times 15$ matrices with $T = 50$, we need to compute FM 3000 times, even for just 10 epochs. Next, we will develop a formulation to make this wFM computation faster since it is invoked hundreds of times in a typical training procedure.

## 4 An efficient way to compute the wFM on $\mathsf{SPD}(n)$

The foregoing discussion describes how the computation of wFM needs an optimization on the SPD manifold. If this sub-module is slow, the demands of the overall runtime will rule out practical adoption. In contrast, if this sub-module is fast but numerically or statistically unstable, the errors will propagate in unpredictable ways, and can adversely affect the parameter estimation. Thus, we need a scheme that balances performance and efficiency.

Estimation of the FM from samples is a well researched topic. For instance, the authors in [45, 49] used Riemannian gradient descent to compute the FM. But the algorithm has a runtime complexity of $\mathcal{O}(iN)$, where $N$ is the number of samples and $i$ is the number of iterations for convergence. This procedure comes with provable consistency guarantees – thus, while it will serve our goals in theory, we find that the runtime for each run makes training incredibly slow. On the other hand, the $\mathcal{O}(N)$ recursive FM estimator using the Stein metric presented in [52] is fast and apt for this task if no additional assumptions are made. However, it comes with no theoretical guarantees of consistency.

**Key Observation.** We found that with a few important changes to the idea described in [52], one can derive an FM estimator that retains the attractive efficiency behavior and is provably consistent. The key ingredient here involves using a novel isometric mapping from the SPD manifold to the unit Hilbert sphere. Next, we present the main idea followed by the analysis.

**Proposed Idea.** Let $\{X_i\}_{i=1}^N \subset \mathsf{SPD}(n)$ for which we want to compute the FM which will be used in (6)–(9). Authors in [52] presented a recursive Stein mean estimator given below:

$$M_1 = X_1 \quad M_k = M_{k-1} \left[ \sqrt{T_k + \frac{(2w_k - 1)^2}{4} (I - T_k)^2} - \frac{2w_k - 1}{2} (I - T_k) \right], \quad (10)$$

where $T_k = M_{k-1}^{-1} X_k$ and $\{w_i\}$ is the set of weights. Instead, briefly, our strategy is **(i)** use an isometric mapping from $\mathsf{SPD}(n)$ to the unit Hilbert sphere; **(ii)** make use of an efficient way to

compute the FM on the unit Hilbert sphere; This isometric mapping to the Hilbert sphere then transfers the problem of proving consistency of the estimator from $\mathsf{SPD}(n)$ to that on the Hilbert sphere, which is easier to prove as shown below. This then leads to consistency of FM estimator on $\mathsf{SPD}(n)$.

We define the isometric mapping from $\mathsf{SPD}(n)$ with a Stein metric to $\mathbf{S}^\infty$, i.e., the infinite dimensional unit hypersphere. In order to define it, notice that we need to define a metric, $d_S$ on $\mathbf{S}^\infty$ such that, $(\mathsf{SPD}(n), d)$ and $(\mathbf{S}^\infty, d_S)$ are *isometric*. This procedure and the associated consistency analysis is described below (all proofs are in the supplement).

**Definition 1.** *Let $A \in \mathsf{SPD}(n)$. Let $f := \mathcal{G}(A)$ be the Gaussian density with $\mathbf{0}$ mean and covariance matrix $A$. Now, we normalize the density $f$ by $f \mapsto f/\|f\|$ to map it onto $\mathbf{S}^\infty$. Let, $\Phi : \mathsf{SPD}(n) \to \mathbf{S}^\infty$ be that mapping. We define the metric on $\mathbf{S}^\infty$ as $d_S(\widetilde{f}, \widetilde{g}) = \sqrt{-\log\langle \widetilde{f}, \widetilde{g}\rangle^2}$.*

Here, $\langle , \rangle$ is the $L^2$ inner product. The following proposition proves the isometry between $\mathsf{SPD}(n)$ with the Stein metric and the hypersphere with the new metric. Let, $A, B \in \mathsf{SPD}(n)$. Then,

**Proposition 1.** *Let $\widetilde{f} = \Phi(A)$ and $\widetilde{g} = \Phi(B)$. Then, $d(2A, 2B) = d_S(\widetilde{f}, \widetilde{g})$.*

Note that, $\Phi$ maps a point on $\mathsf{SPD}(n)$ to the positive orthant of $\mathbf{S}^\infty$, denoted by $\mathcal{H}$ since the components of any probability vector are non-negative. We should point out that in this metric space, there are no geodesics since it is *not* a length space. As a result, we cannot simply use the consistency proof of the stochastic gradient descent based FM estimator presented in [10] for any Riemannian manifold and apply it here. Hence, the recursive FM presented next for the identity in (10) with the mapping described above will need a separate consistency analysis.

**Recursive Fréchet mean algorithm on $(\mathcal{H}, d_S)$.** Let $\{\mathbf{x}_i\}_{i=1}^{N}$ be the samples on $(\mathcal{H}, d_S)$ where $\mathcal{H}$ gives the positive orthant of $\mathbf{S}^\infty$. Then, the FM of the given samples, denoted by $\mathbf{m}^*$, is defined as $\mathbf{m}^* = \arg\min_{\mathbf{m}} \sum_{i=1}^{N} d_S^2(\mathbf{x}_i, \mathbf{m})$. Our recursive algorithm to compute the wFM of $\{\mathbf{x}_i\}_{i=1}^{N}$ is:

$$\mathbf{m}_1 = \mathbf{x}_1 \quad \mathbf{m}_k = \arg\min_{\mathbf{x}} \left( w_k\, d^2(\mathbf{x}_k, \mathbf{x}) + (1 - w_k)\, d^2(\mathbf{m}_{k-1}, \mathbf{x}) \right) \tag{11}$$

where, $\mathbf{m}_k$ is the $k^{th}$ estimate of the FM. At each step of our algorithm, we simply calculate a wFM of two points and we chose the weights to be the Euclidean weights. So, in order to construct a recursive algorithm, we need to have a closed form expression of the wFM, as stated next.

**Proposition 2.** *The minimizer of (11) is given by $\mathbf{m}_k = \frac{\sin(\theta-\alpha)}{\sin(\theta)}\mathbf{m}_{k-1} + \frac{\sin(\alpha)}{\sin(\theta)}\mathbf{x}_k$, where $\theta = \arccos(\langle\mathbf{m}_{k-1}, \mathbf{x}_k\rangle)$ and $\alpha = \arctan\left( \frac{-1+\sqrt{4c^2(1-w_k)-4c^2(1-w_k)^2+1}}{2c(1-w_k)} \right)$ and $c = \tan(\theta)$.*

**Consistency and Convergence analysis of the estimator.** The following proposition (see supplement for proof) gives us the weak consistency of this estimator and also the convergence rate.

**Proposition 3.** *(a) Var $(\mathbf{m}_k) \to 0$ as $k \to \infty$. (b) The rate of convergence of the proposed recursive FM estimator is super linear.*

Due to proposition 1, we obtain a consistency result for (10) with our mapping. These results suggest that we now have a suitable FM estimator which is **consistent and efficient** – this can be used as a black-box module in our RNN formulation in (6)-(9).

## 5 Experiments

In this section, we demonstrate the application of SPD-SRU to answer three key questions **(1)** Using the manifold constraint, what are we saving in terms of number of parameters/ time and is the performance competitive? **(2)** When data is not manifold valued, can we use our framework with the geometry constraint? **(3)** In a real application, what improvements can we get over the baseline? We perform three sets of experiments to answer these questions namely: (a) classification of moving patterns on Moving MNIST data, (b) classification of actions on UCF11 dataset and (c) permutation testing to detect group differences between patients with and without Parkinson's disease. In the following subsections, we discuss about each of these dataset in more detail and present the performance of our SPD-SRU. Our code is available from `https://goo.gl/SfAezS`.

### 5.1 Savings in terms of number of parameters/ time and experiments on vision datasets.

In this section, we perform two sets of experiments namely (1) classification of moving patterns on Moving MNIST data, (2) classification of actions on UCF11 data to show the improvement of our proposed framework over the state-of-the-art methods in terms of number of parameters/ time. We compared with LSTM [28], SRU [47], TT-GRU and TT-LSTM [66]. In the first two

classification applications, we use a convolution block before the recurrent unit for all the competitive methods except for TT-GRU and TT-LSTM. In our SPD-SRU model, before the recurrent layer, we included a covariance block analogous to [67] after one convolution layer ([67] includes details of the construction for the covariance block). So, the input of our SPD-SRU layer is a sequence of matrices in $\mathsf{SPD}(c+1)$, where $c$ is the number of channels from the convolution layer.

**Classification of moving patterns in Moving MNIST data.** We used the Moving MNIST data as generated in [57]. For this experiment we performed 2 and 3 classes classification experiment. In each class, we generated 1000 sequences each of length 20 showing 2 digits moving in a $64 \times 64$ frame. Though within a class, the digits are random, we fixed the moving pattern by fixing the speed and direction of the movement. In this experiment, we kept the speed to be the same for all the sequences, but two sequences from two different classes can differ in orientation by at least $5°$ and by at most $30°$. We experimentally see that, SPD-SRU can achieve very good 10-fold testing accuracy even when the orientation difference of two classes is $5°$. In fact SPD-SRU uses the smallest number of parameters among all methods tested and still offers the best average testing accuracy.

In Table 1, we report the mean and standard deviation of the 10-fold testing accuracy. We should point out that the training accuracy for all the competitive methods is $> 95\%$ for all cases. For TT-RNN, we reshaped the input to be $4 \times 8 \times 8 \times 16$ and kept the

| Mode | # params. | time (s) / epoch | orientation (°) | | |
|---|---|---|---|---|---|
| | | | 30-60 | 10-15 | 10-15-20 |
| SPD-SRU | **1559** | $\sim 6.2$ | **1.00 ± 0.00** | **0.96 ± 0.02** | **0.94 ± 0.02** |
| TT-GRU | 2240 | $\sim$ **2.0** | **1.00 ± 0.00** | 0.52 ± 0.04 | 0.47 ± 0.03 |
| TT-LSTM | 2304 | $\sim$ **2.0** | **1.00 ± 0.00** | 0.51 ± 0.04 | 0.37 ± 0.02 |
| SRU | 159862 | $\sim 3.5$ | **1.00 ± 000** | 0.75 ± 0.19 | 0.73 ± 0.14 |
| LSTM | 252342 | $\sim 4.5$ | 0.97 ± 0.01 | 0.71 ± 0.07 | 0.57 ± 0.13 |

Table 1: Comparative results on Moving MNIST

output shape and rank to be $4 \times 4 \times 4 \times 4$ and $1 \times 4 \times 4 \times 4 \times 1$. The number of output units for LSTM is set to 10 and the number of statistics for SRU is set to 80. Note that, we chose different parameters for SRU and LSTM and TT-RNN and the one we report here are those for which the number of parameters are smallest for the reported testing accuracy. For the convolution layer, we chose the kernel size to be $5 \times 5$ and the input and output channels to be 5 and 10 respectively, i.e., the dimension of the SPD matrix is 11 for this experiment. As before, the parameters are chosen so the number of parameters are smallest to get the reported testing accuracy.

One can see from the table that, SPD-SRU takes the least number of parameters and can achieve very good classification accuracy even for $5°$ orientation difference and for three classes. Note that TT-RNN is the closest to SPD-SRU in terms of parameters. For comparisons, we conduct an experiment where we vary the difference of orientation from $30°$ to $5°$. The testing accuracies are shown in Fig. 2. We can see that only SPD-SRU maintains good 10-fold testing accuracy for all orientation differences while the performance of TT-RNN (both variants) deteriorates as we decrease the difference between orientations of the two classes

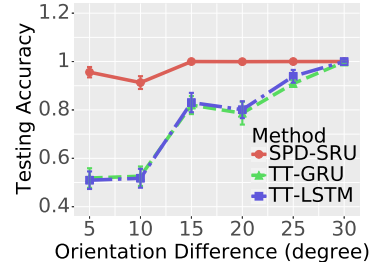

Figure 2: Comparison of testing accuracies with varying orientations

(the effect size). In terms of training time, SPD-SRU takes around 6 seconds per epoch while the fastest method is TT-RNN which takes around 2 seconds. But, in this experiment, SPD-SRU takes 75 epochs to converge to the reported results while TT-RNN takes around 400 epochs. So, although TT-RNN is faster per epoch, the total training time for TT-RNN and SPD-SRU is almost the same. We also should point out that although the number of trainable parameters are fewer for SPD-SRU than TT-RNN, the time difference is due to constructing the covariance in each epoch which can be optimized via faster implementations.

**Classification of moving patterns in UCF-11 data.** We performed an action classification experiment on UCF11 dataset [43]. It contains in total 1600 video clips belonging to 11 classes that summarize the human action visible in each video clip such as basketball shooting, diving and others. We followed the same processing step as in [66]. Each frame has resolution $320 \times 240$. We generate a sequence of RGB frames of size $160 \times 120$ from each clip at 24 fps. The lengths of frame sequences from each video therefore are in the range of 204-1492 with an average of 483.7. For SPD-SRU, we chose two convolution layers with kernel size $7 \times 7$ and number of output channels to be 5 and 7 respectively and then 5 PSRN layers. Hence, the dimension of the covariance matrices are $8 \times 8$ for this experiment. For TT-GRU and TT-LSTM, we used the same configurations of input and output factorization as given in [66]. For SRU and LSTM we used the number of statistics and number of output units to be 750. For both SRU and LSTM we used 3 convolution layers with kernel size $7 \times 7$ and output channels to be 10, 15 and 25 respectively to get the reported testing accuracies.

All the models achieve $> 90\%$ training accuracy. We report the testing accuracy with the number of parameters and time per epoch in Table 2. From this experiment, we can see that the number of parameters for SPD-SRU is significantly smaller than the other models without sacrificing the testing accuracy. In terms of training time, SPD-SRU takes approximately 3 times more time than TT-RNN but SPD-SRU (TT-RNN) converges in 50 (100) epochs. Furthermore, we like to point out that after 400 epochs, SPD-SRU gives $79.90\%$ testing accuracy. Hence, analogous to the previous experiment, we can conclude that SPD-SRU maintains very good classification accuracy while keeping the number of trainable parameters very small. Furthermore, this experiment indicates that SPD-SRU can achieve competitive performance on real data with small number of training parameters in comparable time.

## 5.2 Application on manifold valued data

From the previous two experiments, we can conclude that SPD-SRU requires a smaller number of parameters. Now, we focus our attention to a neuroimaging application where data is manifold valued. Because the number of parameters are small, we can do statistical testing on brain connectivity at the fiber bundle level. We seek to find group differences between subjects with and without Parkinson's disease (denoted by 'PD' and 'CON') based on the M1 fiber tracts on both hemispheres of the brain.

**Permutation testing to detect group differences.** The data pool consists of dMRI (human) brain scans acquired from 50 'PD' patients and 44 'CON' healthy controls. All images were collected using a 3.0T MR scanner (Philips Achieva) and 32-channel quadrature volume head coil. The parameters of the diffusion imaging acquisition sequence

| Model | # params. | time/ epoch | Test acc. |
|---|---|---|---|
| SPD-SRU | **3337** | $\sim 76$ | **0.78** |
| TT-GRU | 6048 | $\sim 42$ | **0.78** |
| TT-LSTM | 6176 | $\sim$ **33** | **0.78** |
| SRU | 2535630 | $\sim 50$ | 0.75 |
| LSTM | 14626425 | $\sim 57$ | 0.70 |

Table 2: Comparative results on UCF11 data

were: gradient directions = 64, b-values = 0/1000 s/mm2, repetition time =7748 ms, echo time = 86 ms, flip angle = $90°$, field of view = $224 \times 224$ mm, matrix size = $112 \times 112$, number of contiguous axial slices = 60 and SENSE factor P = 2. We used FSL [8] software to extract M1 fiber tracts (denoted by 'LM1' and 'RM1') [5], which consists of 33 and 34 points respectively (please see Fig. 3 for M1-SMATT fiber tract template). We fit a diffusion tensor and extract $3 \times 3$ SPD matrices. Now, for each of these two classes, we use 3 layers of SPD-SRU to learn the tracts pattern to get two models for 'PD' and 'CON' (denoted by 'mPD' and 'mCON').

Now, we use a permutation testing based on a "distance" between 'mPD' and 'mCON'. We will define the distance between two network models as proposed in [59] (let it be denoted by $d_{\text{mod}}$). Here, we assume each subject is independent hence use of permutation testing is sensible. Then we perform permutation testing for each tract as follows (i) randomly permute the class labels of the subjects and learn 'mPD' and 'mCON' models for each of the new group. (ii) compute $d_{\text{mod}}^j$ (iii) repeat step (ii) 10,000 times and report the $p$-value as the fraction of times $d_{\text{mod}}^j > d_{\text{mod}}$. So, we ask if we can reject the null hypothesis that *there is no significant difference between the tracts models learned from the two different classes*. As a baseline, we use the following scheme: (i) for each tract of each subject, compute the FM of the matrices on the tract. (ii) use Cramer's test based on this Stein distance. (iii) do the permutation testing based on the Cramer's test.

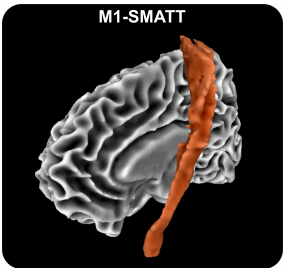

Figure 3: M1-SMATT template

We found that using our SPD-SRU model with 3 layers, the $p$-value for 'LM1' and 'RM1' are 0.01 and 0.032 respectively, while the baseline method gives a p-value of 0.17 and 0.34 respectively. Hence, we conclude that, unlike the baseline method, using SPD-SRU we can reject the null hypothesis with $95\%$ confidence. To the best of our knowledge, this is the first result that demonstrates a RNN based statistical significance test applied on tract based group testing in neuroimaging.

## 6 Conclusions

Non-Euclidean or manifold valued data are ubiquitous in science and engineering. In this work, we study the setting where the data (or measurements) are ordered, longitudinal or temporal in nature and live on a Riemannian manifold. This setting is common in a variety of problems in statistical machine learning, vision and medical imaging. We presented a generalization of the RNN to such non-Euclidean spaces and analyze its theoretical properties. Our proposed framework is fast and needs far fewer parameters than the state-of-the-art. Experiments show competitive performance on benchmark computer vision datasets in comparable time. We also apply our framework to perform statistical analysis in brain connectivity and demonstrate the applicability to manifold valued data.

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
