[Supplementary Material · supplement.pdf]

# Statistical Recurrent Network on the space of symmetric positive definite matrices (Supplementary material)

## 1 Proofs for the propositions stated in the paper

**Proposition 1.** $\Phi$ *as defined in the paper is a bijection from* $\mathsf{SPD}(n)$ *onto* $\Phi\left(\mathsf{SPD}(n)\right)$.

*Proof.* Let $\Phi = \Psi \circ \beta$ where $\Psi : \mathsf{SPD}(n) \to \mathcal{N}$ and $\beta : \mathcal{N} \to \mathbf{S}^\infty$, where $\mathcal{N}$ is the space of zero mean $n$-variate Gaussian densities. Let $\mathcal{H} = \beta\left(\mathcal{N}\right)$. Clearly $\Psi$ is invertible, since, given a zero-mean Gaussian distribution, we can get its covariance matrix. One can think of this as drawing samples from the density and the sample covariance matrix asymptotically converges to the covariance of the Gaussian distribution. Now, given $\widetilde{f} \in \mathcal{H}$, $f = \frac{\widetilde{f}}{\int \widetilde{f}}$ such that, $\widetilde{f} = \beta(f)$. Thus, we can prove that $\Phi = \Psi \circ \beta$ is a bijection from $\mathsf{SPD}(n)$ onto $\Phi(\mathsf{SPD}(n))$. $\qquad\square$

**Proposition 2.** *Let,* $A, B \in \mathsf{SPD}(n)$. *Let* $\widetilde{f} = \Phi\left(A\right)$ *and* $\widetilde{g} = \Phi\left(B\right)$. *Then,* $d\left(2A, 2B\right) = d_S\left(\widetilde{f}, \widetilde{g}\right)$.

*Proof.*

$$d_S\left(\widetilde{f}, \widetilde{g}\right) = \sqrt{-\log\langle \widetilde{f}, \widetilde{g}\rangle^2}$$

$$= \sqrt{-2\ \log\left(\frac{\langle f, g\rangle}{\|f\|\|g\|}\right)}$$

$$= \sqrt{-2\log\left(\frac{\left((2\pi)^3 \det\left(A + B\right)\right)^{-1/2}}{\left((2\pi)^3 \det\left(2A\right)\right)^{-1/4}\left((2\pi)^3 \det\left(2B\right)\right)^{-1/4}}\right)}$$

$$= \sqrt{-2\left[\frac{-\log\det\left(A + B\right)}{2} + \frac{\log\det\left(2A\right)}{4} + \frac{\log\det\left(2B\right)}{4}\right]}$$

$$= \sqrt{\log\det\left(A + B\right) - 1/2\log\det\left(2A\right) - 1/2\log\det\left(2B\right)}$$

$$= d\left(2A, 2B\right)$$

In the above proof, we have used the fact that, $\langle f, g\rangle = \left((2\pi)^3 \det\left(A + B\right)\right)^{-1/2}$, where $f$ and $g$ are zero-mean Gaussian densities with covariances $A$ and $B$ respectively. $\qquad\square$

**Proposition 3.** $(\mathcal{H}, d_S)$ *is a compact and complete metric space but not a length space.*

---

14 *Proof.* The symmetry, non-negativity and the identity of the indiscernible are easy to prove. In order
15 to prove triangle inequality, observe that $I - \mathbf{y}\mathbf{y}^t$ is positive semi-definite, for all $\mathbf{y} \in \mathcal{H}$. As, $\mathbf{x}, \mathbf{z} \in \mathcal{H}$,
16 $\langle \mathbf{x}, \mathbf{y} \rangle \langle \mathbf{y}, \mathbf{z} \rangle \leq \langle \mathbf{x}, \mathbf{z} \rangle$. Now, since $\log$ is an increasing function, we get $d_S(\mathbf{x}, \mathbf{y}) + d_S(\mathbf{y}, \mathbf{z}) \geq d(\mathbf{x}, \mathbf{z})$.
17 This proves that $(\mathcal{H}, d_S)$ is a metric space.

18 Since any compact metric space is complete, it suffices to show that $(\mathcal{H}, d_S)$ is a compact metric
19 space. Let $\Gamma : (\mathcal{H}, d_A) \to (\mathcal{H}, d_S)$, where $d_A$ is the arc-length metric restricted to $\mathcal{H}$. Then,
20 $\Gamma(x) = \sqrt{-\log \cos^2(x)}$. Hence, $\frac{d\Gamma}{dx} = \frac{\tan(x)}{\sqrt{-\log \cos^2(x/2)}}$. Since $x \in [0, \pi/2)$, $\Gamma$ is an increasing
21 function. Now, let $\epsilon > 0$ and let $\mathbf{y} \in \mathcal{H}$, and $d_A(\mathbf{x}, \mathbf{y}) \leq \epsilon$ implies, $d_S(\mathbf{x}, \mathbf{y}) \leq \Gamma(\epsilon) > 0$. Now,
22 choose $\delta = \Gamma(\epsilon)$ to conclude that $\Gamma$ is continuous and as $(\mathcal{H}, d_A)$ is compact so is $(\mathcal{H}, d_S)$. $\qquad \square$

23 **Proposition 4.** *The minimizer of Eq. 14 (in the paper) is given by* $\mathbf{m}_k = \frac{\sin(\theta - \alpha)}{\sin(\theta)} \mathbf{m}_{k-1} + \frac{\sin(\alpha)}{\sin(\theta)} \mathbf{x}_k,$

24 *where* $\theta = \arccos(\langle \mathbf{m}_{k-1}, \mathbf{x}_k \rangle)$ *and* $\alpha = \arctan\left( \frac{-1 + \sqrt{4c^2(1-w_k) - 4c^2(1-w_k)^2 + 1}}{2c(1-w_k)} \right)$ *and* $c =$

25 $\tan(\theta)$.

26 *Proof.* Let $\alpha = \arccos(\langle \mathbf{m}_{k-1}, \mathbf{m}_k \rangle)$. Let, $\theta = \arccos(\langle \mathbf{m}_{k-1}, \mathbf{x}_k \rangle)$. Define,
$$g(\alpha) = -(1 - w_k) \log(\cos^2(\alpha)) - (w_k) \log(\cos^2(\theta - \alpha))$$
27 Then, the partial of $g(\alpha)$ with respect to $\alpha$ is given by:
$$\frac{\partial g(\alpha)}{\partial \alpha} = 2(1 - w_k) \tan(\alpha) - 2w_k \tan(\theta - \alpha)$$
28 After setting $\frac{\partial g(\alpha)}{\partial \alpha} = 0$, we get,
$$\frac{\tan(\alpha)}{\tan(\theta - \alpha)} = \frac{w_k}{1 - w_k}$$
$$\frac{(1 + \tan(\theta)\tan(\alpha))\tan(\alpha)}{\tan(\theta) - \tan(\alpha)} = \frac{w_k}{1 - w_k}$$
29 Let, $x = \tan(x)$, $c = \tan(\theta)$. Thus, we get
$$(1 + cx)x = \frac{w_k}{1 - w_k}(c - x)$$
$$cx^2 + (1 + \frac{w_k}{1 - w_k})x - c\frac{w_k}{1 - w_k} = 0$$
$$cx^2 + \frac{1}{1 - w_k}x - c\frac{w_k}{1 - w_k} = 0$$

30 Solving the above quadratic, we get
$$x = \frac{-1 + \sqrt{4c^2(1 - w_k) - 4c^2(1 - w_k)^2 + 1}}{2c(1 - w_k)}$$
$$\alpha = \arctan\left( \frac{-1 + \sqrt{4c^2(1 - w_k) - 4c^2(1 - w_k)^2 + 1}}{2c(1 - w_k)} \right)$$
31 Now, as $\alpha = \arccos(\langle \mathbf{m}_{k-1}, \mathbf{m}_k \rangle)$, $\mathbf{m}_k = \frac{\sin(\theta - \alpha)}{\sin(\theta)} \mathbf{m}_{k-1} + \frac{\sin(\alpha)}{\sin(\theta)} \mathbf{x}_k$. $\qquad \square$

32 **Proposition 5.** *Var* $(\mathbf{m}_k) \to 0$ *as* $k \to \infty$.

33 *Proof.* Let $\theta_k = \cos^{-1}\left( \mathbf{m}_k^T \mathbf{m}^* \right) = d_A(\mathbf{m}_k, \mathbf{m}^*)$, then by Taylor's expansion,
$$d^2(\mathbf{m}_k, \mathbf{m}^*) = -\log \cos^2 \theta_k$$
$$= -2 \log \cos \theta_k$$
$$= -2 \left[ -\frac{\theta_k^2}{2} - \frac{\theta_k^4}{12} + O\left(\theta_k^6\right) \right]$$
$$= \theta_k^2 + \frac{\theta_k^4}{6} + O\left(\theta_k^6\right)$$

34 So

$$\lim_{k\to\infty} E\left[d^2\left(\mathbf{m}_k,\mathbf{m}^*\right)\right] = \lim_{k\to\infty} E\left[\theta_k^2\right] + E\left[\frac{\theta_k^4}{6}\right] + E\left[O\left(\theta_k^6\right)\right].$$

35 Since $E\left[\theta_k^2\right] = E\left[d_A^2\left(\mathbf{m}_k,\mathbf{m}^*\right)\right] \to 0$ [3], by dominated convergence theorem and the fact that
36 $\theta_k \in \left[0,\frac{\pi}{2}\right]$, $E\left[\lim_{k\to\infty}\theta_k^2\right] = \lim_{k\to\infty} E\left[\theta_k^2\right] = 0$. So $\lim_{k\to\infty}\theta_k^2 = 0$ $(\because \theta_k^2 \geq 0)$. Then again
37 by dominated convergence theorem, $\lim_{k\to\infty} E\left(\theta_k^{2n}\right) = E\left[\lim_{k\to\infty}\theta_k^{2n}\right] = 0$ for $n \in \mathbb{N}$. Thus

$$\lim_{k\to\infty} Var\left(\mathbf{m}_k\right) = \lim_{k\to\infty} E\left[d^2\left(\mathbf{m}_k,\mathbf{m}^*\right)\right] = 0$$

38 $\hfill\square$

39 **Proposition 6.** *The rate of converge of the proposed recursive FM estimator is super linear.*

*Proof.*

$$d(\mathbf{m}_n,\mathbf{m}_m) \leq d(\mathbf{m}_n,\mathbf{m}_{n+1}) + \cdots + d(\mathbf{m}_{m-1},\mathbf{m}_m)$$

$$= \sqrt{-2\log\cos\alpha_{n+1}} + \cdots + \sqrt{-2\log\cos\alpha_m}$$

$$= \sum_{k=n+1}^{m}\sqrt{-2\log\cos\alpha_k}$$

$$\leq (m-n-1)\sqrt{-\frac{2}{m-n-1}\sum_{k=n+1}^{m}\log\cos\alpha_k}$$

$$= \sqrt{-2(m-n-1)\log\left(\prod_{k=n+1}^{m}\cos\alpha_k\right)}$$

40 where, $\alpha_n = \tan^{-1}\left(\frac{-1+\sqrt{4c^2\left(1-\frac{1}{n}\right)-4c^2\left(1-\frac{1}{n}\right)^2+1}}{2c\left(1-\frac{1}{n}\right)}\right)$, where $c = \tan(\theta_n)$ and $\theta_n =$
41 $\cos^{-1}\mathbf{m}_{n-1}^t\mathbf{x}_n$. Now, we have

$$\tan\alpha_n = \frac{-1+\sqrt{4c^2\left(1-\frac{1}{n}\right)-4c^2\left(1-\frac{1}{n}\right)^2+1}}{2c\left(1-\frac{1}{n}\right)}$$

$$= \frac{-\frac{n}{n-1}+\sqrt{\frac{4c^2 n}{(n-1)}-4c^2+\frac{n^2}{(n-1)^2}}}{2c}$$

$$\approx \frac{-\frac{n}{n-1}+1+\frac{1}{2}\left[\frac{4c^2 n}{(n-1)}-4c^2+\frac{n^2}{(n-1)^2}-1\right]}{2c}$$

$$= \frac{\frac{1}{2}\left[\frac{n}{n-1}-1\right]^2+\frac{1}{2}\left[\frac{4c^2 n}{(n-1)}-4c^2\right]}{2c}$$

$$\approx -\frac{1}{2}\frac{1}{(n-1)^2}+\frac{2c^2}{n-1}$$

42 Using the taylor series of $\tan^{-1}(x)$,

$$\alpha_n \approx \tan^{-1}\left(-\frac{1}{2}\frac{1}{(n-1)^2}+\frac{2c^2}{n-1}\right)$$

$$= -\frac{1}{2}\frac{1}{(n-1)^2}+\frac{2c^2}{n-1}+O\left(\left(\frac{1}{n-1}\right)^6\right)$$

43 Hence, $\frac{1}{n^2} < \alpha_n < \frac{1}{n}$. It is easy to show using the proof in [1], that for $\alpha_n = \frac{1}{n}$ we get a linear
44 convergence rate. Hence, the rate of convergence is super-linear.

45 $\hfill\square$

## 1.1 Discretization of $\Phi$

Given $A \in \mathsf{SPD}(n)$, we have a mapping $\Phi : \mathsf{SPD}(n) \to \mathbf{S}^\infty$ that maps $A \mapsto f/\|f\|$, where $f$ is the Gaussian density of zero mean and covariance $A$. Though, this is a well-defined mapping, in experiments we need a discretization of $f/\|f\|$. Given $f \in \mathcal{N}$, we will use Algorithm 1 to get $\tilde{\beta}(f)$ on a finite dimensional manifold.

---

**Algorithm 1:** Algorithm to map a Gaussian density on to a finite dimensional oblique manifold.

---

**Input**: $f \in \mathcal{N}$; $\epsilon > 0$; $b$ : number of bins
**Output**: $\tilde{f} \in \underbrace{\mathbf{S}^{b-1} \times \cdots \times \mathbf{S}^{b-1}}_{n^2 \text{ times.}}$

1 Choose the vector $\mathbf{v}$ uniformly at random from $\mathbf{S}^{n-1}$;
2 For $i = 1, \cdots, n$ and $j = 1, \cdots, n$, let $\mathbf{v}_{ij} = \mathbf{v} + \epsilon(\mathbf{e}_i + \mathbf{e}_j)$, where $\{\mathbf{e}_i\}$ are the canonical basis of $\mathbf{R}^n$;
3 Project $f$ on $\mathbf{v}_{ij}$ to get an univariate zero-mean Gussian of variance $\sigma_{ij}^2$, for each $i, j$;
4 Take a uniform $b$ number of grids in $[-2\sigma_{ij}, 2\sigma_{ij}]$ and get a probability vector for each univariate Gaussian;
5 Use the square root parametrization to map each discretized probability vector on $\mathbf{S}^{b-1}$;
6 Arrange the probability vectors to get a point on the oblique manifold, $\mathbf{S}^{b-1} \times \cdots \times \mathbf{S}^{b-1}$.

---

In Algorithm 1, we have used the projection idea proposed in [2]. We have chosen the interval of discretization as the $95\%$ confidence interval. Here we chose $n^2$ random projections but one may want to use more number of random projections to get a higher resolution.

In the next section, we perform synthetic experiments to demonstrate comparison results of the proposed metric on the hypersphere and Stein metric on $\mathsf{SPD}(n)$ in terms of error and computational time. Furthermore, we demonstrate the efficiency of our recursive FM estimator over it's batch-mode counterpart.

# 2   Synthetic experiments

In this subsection, we performed experiments on synthetic data to show the performance comparison for computing the FM of the data points on the hypersphere (mapped from $\mathsf{SPD}(n)$ to the hypersphere using $\Phi$) endowed with our new metric against the FM of data points on $\mathsf{SPD}(n)$ (prior to the mapping) endowed with the Stein metric. In the latter case, the FM is computed using the recursive FM estimator defined in [4]. Here, we have randomly generated data samples on $\mathsf{SPD}(n)$ from a Log-Normal distribution with mean $I$ and variance $1.0$. We vary the number of samples, $N$ as well as the dimension $n$. For each instance, we compute the FM using both the metrics and plot the performance curves. In the context of the required computation time, we can see that the proposed metric on hypersphere is significantly faster than when using the Stein metric on $\mathsf{SPD}(n)$ as depicted in Fig. 1. In terms of the accuracy of the computed FM with respect to the ground truth FM (which is known for the synthetic data), using the Stein and the proposed metric respectively, we get almost similar variance of the FM estimator. Because of the proven isometry, any difference in the variance is due to the discretization of the density corresponding to the sample SPD matrix on $\mathsf{SPD}(n)$. Though, we have used the discretization as proposed in Algorithm 1, as pointed out earlier, to achieve a better accuracy (smaller error), one may want to use finer discretization. Increased resolution in the discretization will not be of much concern since the new metric is computable much more efficiently than the Stein metric. In our experiments, we observed that even by taking just $n^2$ random projections, we were able to achieve comparable error. Note that, both by varying $n$ and $N$, we can empirically see that computation using the new metric is significantly faster compared to using the Stein metric based FM estimator. Furthermore, in Fig. 1, we present a comparison of the two metrics for computing the FM of samples, depicting the number of samples required by the respective FM estimators to achieve an accuracy within prespecified tolerance. This analysis is required for finite samples and it is evident from the figure that using both of these metrics we need almost same number of samples to achieve the desired error tolerance.

We have also performed experiments to compare the performance of our proposed recursive FM estimator and it's batchmode counterpart. For the gradient descent based batchmode technique, we have used a "warm restart", i.e., we initialize the gradient descent algorithm with the old mean whenever a new sample data point is input to the algorithm. From Fig. 2, we can see that the recursive

Figure 1: Comparison of FM computation time using the Stein and proposed metric.

Figure 2: Comparison between the recursive and batch mode FM estimators

technique is much faster without sacrificing much error. In fact the error from both the estimators are very close but computationally the recursive FM estimator is significantly faster. Further, we also present a time and error/accuracy trade-off plot for the proposed recursive FM estimator. From this plot, we can conclude that the product of time and error/accuracy is bounded from above, which basically indicates that even if the desired error is very small (high accuracy) we need finite number of samples to achieve this.