[Reviews · NeurIPS 2018]

Reviewer 1



The submission #5332, entitled "Statistical Recurrent Models on Manifold valued Data", presents a framework for the fit of Recurrent Neural networks on SPD matrices. Much of the efforts are spent on deriving the framework for these computations. Particular attention is paid to the estimation of Fréchet mean for this type of data, in order to obtain a fast yet reliable estimators. the the authors consider frameworks where data are motion sequences, the modeling of which requires such SPD matrices, or diffusion MRI, where so-called diffusion tensors are manipulated. All these data are very low-dimensional, so that the reported algorithmic optimization unlikely to be useful in practical settings. The experiments show that the proposed approach is accurate in the problems considered and requires fewer iterations for convergence. My main impression is that the paper introduces such sophistication because it is cool, even in the absence of compelling needs. The value of the contribution is thus confined to the computational and algorithmic developments The proposed approach seems rather elegant, but it certainly scales poorly with data dimensionality. For instance, taking Cholesky factor has a cubic complexity, which is ignored in the present context where the dimensionalities are small. The derivations seem sound. Personally, I do not see the point is dealing with deep learning as an aesthetic game, and am rather skeptical about the value of this contribution. The experiments are not strongly motivated. For instance in neuroimaging, people stopped dealing with diffusion tensors about 10 years ago. Developing in 2018 a testing framework for this type data seems anecdotal at best. Some comment on moving mnist. Overall, the experimental framework is thus full of weird choices, and seems in general quite unnatural. It is a pity, because there exist real use cases for SPD matrices in some domains. The paper is somewhat hard to understand. The authors overemphasize some details and do not give the big picture. I had a hard time figuring out what is x_t in eq (3), and o_t in eq (4). One needs to read further to understand that. Finally, the paper is unpleasant to read because of overlapping lines. The authors have played with negative \vspaces, which is simply bad.

Reviewer 2



Overall, I found this work interesting and the direction is good. The paper, especially the introduction, is well written and most of the arguments are reasonable. If my concerns can be reasonably addressed, this could be considered as a paper. Summary of the work: Geometric perspective and theoretical understanding are both increasingly important in the deep learning field. Since 2017, there has been a lot of interests in a subfield called geometric deep learning. In this work, the authors first identified that: previous works have been focusing on applying these geometric ideas to CNN trained with functions (data) defined on manifold and graphs, there has been a lack of work on applying this geometric perspective to RNN. Then author applied this geometric perspective to a specific type of RNN models called statistical recurrent units (SRU), which was proposed relatively recently and was trained with data from Euclidean space.With a manifold assumption, the authors need to generalized different operations to respect the geometry. The proposed model achieved both training speeding up and a better performance on several tasks. Comments: 1. A reader may easily find the abstract is too general, which is almost misleading. It doesn't reflect well what the authors actually do in this paper. It promised a general manifold perspective in RNN models, however, all what the authors did was generalizing SRU to SPD matrix manifold. SPD matrices are both interesting and important, but by no means the authors' concrete work can support the claim in the abstract. In fact, for readers familiar with Bronstein et. al. 2017, SPD matrice manifold would not even be the first choice. I highly suggest the authors to make the claim more concrete and reflect what they actually do to make sure the readers' expectation is not set too high. As a result, I was wondering "why SPD?" in the section 2 and then I was wondering "why SRU?" in section 3. In the beginning section 2, the authors claim "... our development is not limited to SPD(n) ...", I haven't found any support. If the authors insist to keep such big claims, please provide at least two types of important and representative manifolds or convey the generality better. 2. Please discuss more about the design principles. In section 2, the authors' choice of the metric and "geometric means" are purely based on the computational cost. A reader may have questions like "What about other properties?". A minimal comparison of different choices is desirable, or please point the interested reader to the right references. Here is a concrete example of the questions: E.g. there are many different metrics for SPD(n), Log Euclidean, Cholesky, power-Euclidean, Affine-invariant, Root Stein Divergence etc. For a computational efficiency, Root Stein Divergence does seem to be a good once. But it doesn't give a geodesic, however, Log Euclidean does. It's very important to discuss these since the actual model proposed in this paper is not very general, the design principles might be more important to the readers so that we can learn something easier applicable to their works. And in return, the impact of the work can be larger. Another reference should be cited in the section 2 to fill some gaps in the references already cited: Sra 2012, A new metric on the manifold of kernel matrices with application to matrix geometric means. 3. In section 3, this is the place where the authors first mention their work is actually to general SRU to SPD matrix manifold. As I mentioned before, this is too late, the authors should discuss it earlier. 4. Also at the end of section 3, it might be good to have a discuss on how to in general generalize the operations in RNN to manifold valued data. 5. What's the unique challenge in applying the geometric deep learning perspective to RNN models? And what new tools should be introduced to take extra care when using this perspective to RNN rather than CNN. A very important insight in geometric deep learning is to assume the signal is a function defined on some non-Euclidean domain, specifically a function defined on Riemannian manifold or an undirected graph. Also a manifold itself is very different from a function defined on a manifold. Update after reading authors' reply: I think the response is reasonable and I slightly increased my evaluation score. But, again, I still can not evaluate the choice of SRU + SPD and further the authors should make an effort to show the applicability of the work to the other RNN models. Also the authors should make the claims more concrete in the final revision.

Reviewer 3



OVERVIEW: At a high level, the paper extends a RNN, the Statistical Recurrent Unit (SRU) to the manifold of Symmetric Positive Definite (SPD) matrices to propose a new RNN called SPD-SRU. The idea is motivated by current trends of Geometric Deep Learning for graphs and manifolds and is of interest to the community. The main contributions of the authors lie in the technical implementation of this idea. They extend the update equations for the SRU (Eqns. 2-5), defined for a Euclidean metric space, to the SPD manifold (Eqns. 6-9). This involves (i) posing the linear or fully connected layer WX, as multiple weighted Frechet Mean (wFM) problems (ii) the bias term of the linear layer is captured a group translation operation and (iii) a ReLU non-linear activation is used in the parameter space. The trick that makes this work efficiently is a new proposed algorithm to compute the wFM efficiently using recursive closed form updates. The novelty of the paper is in the derivation of this computation with proofs of consistency and convergence. They evaluate their proposed SRU-RNN on three experiments: (1) Moving MNIST, (2) UCF11 and (3) permutation testing to detect group differences between patients with and without Parkinson's disease. STRENGTHS: 1. The idea is well motivated and the background material, both Riemannian geometry on the SPD manifold and the SRU is covered nicely. 2. The authors have 65 citations in their submission which is indicative of the depth in which they review the literature. 3. The math is clear at a high level in motivating what they want to do, the technical challenges it involves and their solutions to these challenges. 4. They provide proofs for different mathematical properties of their proposed solution to the wFM problem for SPD matrices. 5. Their experimental evaluation shows that they achieve comparable (or better in some cases) performance for a fraction of the number of parameters to other RNN units like LSTM, SRU, TT-GRU, TT-LSTM. WEAKNESSES: 1. The single biggest critique I have with the submission is with regards to how it is written. In two locations: (1) the proofs provided in the supplementary material and (2) the experimental evaluation, the authors can do a better job at making the paper more accessible. They skip steps or leave parts to the reader which lead to incomplete or confusing proofs. In the experimental evaluation, they provide some details but these details are probably sufficient for the authors to redo their experiments but not for a new researcher trying out their models. If the aim is to make the SPD-SRU as common and popular as the RNN/LSTM, the authors need to make these details more accessible. I agree that page restrictions play a role in how the paper is presented but all these details can and should be provided as part of the supplementary material. I specify the exact problems I refer to above: (i) In line 93, "One can easily see that, with the Stein metric, G is the set of nxn orthogonal matrices, denoted by O(n)". Should it be "a choice of G is the set"? "G is the set" seems to indicate an if and only if statement. The if direction is clear but the only if is not. (ii) All the discussions of network architectures in Sections 5.1 and 5.2 especially with regard to the number of parameters. A table outlining the final chosen network architectures (in the supplementary if needed) with specific choices of layer dimensions is required. This is important to firstly understand how you get the #params column in your tables and secondly to see if the comparison is fair. You want the reader to trust you but you also want him/her to be able to verify these statements made as facts. (iii) Proposition 1: For the \beta function, you define co-domain as S^\infty and range as \mathcal{H} = \beta (\mathcal{N}). The way I see it you have shown that for any \tilde{f} in \mathcal{H}, there exists an f and for any f there exists a \tilde{f}. What I'm missing is the connection between co-domain S^\infty and range \mathcal{H}. I'm also missing a one-to-one mapping from \mathcal{N} to \mathcal{H} to show that many f do not map to a single \tilde{f}. I think a lot of this goes away if you explicitly define what \beta is ? or what the function norm \| f \| is ? (iv) Proposition 2 is correct but Line 11, "we have used the fact that = ..." is left as homework to the reader. It checks out with Linear algebra properties (A^-1 + B^-1)^-1 = A (A+B)^-1 B and det(AB) = det(A) det(B) used in the middle but adding this proof is a couple of extra lines of work that make the proposition complete. A secondary question is why is this proposition needed ? I do not see it explicitly being called in any future proof. (v) Proposition 3: Line 15, I-yy^t for functional y \in \mathcal{H} is not clear. Also, how do you use it in the proof ? How do you get \leq ? (vi) Proposition 7: This is the most important proof of the paper and it's also the most confusing. You start by defining \alpha and \theta. Then you proceed to define g(\alpha) which I assume has some connection to Eqn.11 (not mentioned explicitly and not clear, especially the \log(\cos^2(\theta-\alpha)) term). Then you basically solve for \alpha as a function of \theta and finally write Line 31. The equation for the closed form update in Line 31 checks out but again, it's not clear how you get there. (vii) Proposition 5: Please add the approximations in the Taylor expansion \cos \theta_k = 1 - \frac{\theta_k^2}{2} + \frac{\theta_k^4}{4} + ... and \log(1-x) = -x -\frac{x^2}{2} + ... and the couple of extra steps between equality 2 and 3. (viii) Proposition 6: Not clear what inequality is used in \sum_{k=n+1}^m \sqrt(-2\log \cos \alpha_k) \leq (m-n-1) \sqrt{-\frac{2}{m-n-1} \sum... }. Not clear what happens to the 1/2c term in the approximation above line 42. Most of the math presented here checks out and I trust the authors but complete proofs are important to verify them and for future readers. 2. A key detail that is only briefly mentioned is the implementation of the requirement that the weights are convex \geq 0 and \sum = 1. To me the biggest jump from Eqn.2 to Eqn.6 is this requirement and there is no discussion about changing the Linear layer in this manner, why this is not a sub-optimal step and why this still makes sense. 3. Experimental evaluation: The authors report time taken per epoch during training in Tables 1 and 2 for their experiments on Moving MNIST and UCF11, which is unusual. Standard practice is to report time taken at inference. It is good to be able to train a model in reasonable amount of time but for the time scales they report (a few seconds to maybe 100s per epoch) it does not seem to be a bottleneck. I would be more interested in inference times across models. Also, the permutation testing experiment in Sec.5.2 makes sense but I would propose a modification. As a machine learning practitioner, I would suggest a leave-one-out experiment where you remove one subject from the data, train your with-disease and without-disease models and assign the subject to whichever model it is closest to. This might also be a future diagnostic tool based on your work ? MINOR COMMENTS: 1. Data is singular uncountable ? Lines 1 and 21. 2. Move references to per property in Lines 30-31. 3. Before Sec.2 motivate why SPD is important. I think current literature review is good but it is missing a survey of SPD based models especially deep learning ones. 4. Repetition in Lines 291-295 and Lines 300-305 5. Citation needed for PSRN in Line 354 6. Line 29 of Supplementary, x = \tan(\alpha) DECISION: I have marked the overall rating as 8. Quality: 8 Clarity: 7 Originality: 7 Significance: 8 Update post rebuttal: I have read the author response and other reviews. Based on the response and discussions with other reviewers, I have downgraded by review from 8 to 7. This is due to two reasons: (1) The authors have oversold their claim of generalizing over multiple manifolds. This should either be backed up with technical details or the text modified to reflect the current SPD + SRU setting, which might still be sufficient if convincing arguments are made for it. (2) Scalability is not clear. I am not in complete agreement with R1 that the algorithm is restricted to toy examples but I agree with his/her opinion that this has not been dealt with sufficiently by the authors.